

# The Coupling of a High-efficiency Aerosol Collector with Electrospray Ionisation/Orbitrap Mass Spectrometry as a novel tool for Real-time Chemical Characterisation of Fine and Ultrafine Particles

5  Yik-Sze Lau[1], Zoran Ristovski[1], Branka Miljevic[1]

International Laboratory for Air Quality and Health, School of Earth and Atmospheric Sciences, Queensland University of Technology, Brisbane, Australia

Correspondence to: Branka Miljevic (b.miljevic@qut.edu.au)

**Abstract.** The chemical properties of aerosols in the atmosphere significantly influence their impact on the global climate
forcing and human health. However, a real-time molecular-level characterisation of aerosols remains challenging due to the complex nature of their chemical composition. The current study constructed an instrumental system for the real-time chemical characterisation of aerosol particles. The proposed setup consists of a custom-built high-efficiency aerosol collector (HEAC) used to collect aerosol samples into a working fluid and an electrospray ionisation (ESI) Orbitrap Mass spectrometer (MS) for the subsequent chemical analysis of the liquid sample. The HEAC/ESI-Orbitrap-MS was calibrated against six chemical
compounds to investigate the system's sensitivity and limit of detection (LOD). Results showed that the coupled system has high sensitivities towards the tested chemical compounds and a similar, if not better, LOD than other related instrumental techniques. The capability of the HEAC/ESI-Orbitrap-MS system to identify the chemical composition of organic aerosols (OA) was also examined. Sample OA was generated by α-pinene ozonolysis, and the chemical characterisation results were compared to similar studies. Our data showed that the HEAC/ESI-Orbitrap-MS system can identify most of the α-pinene
ozonolysis products reported in the literature, including cis-pinonic acid, pinalic acid and 3-methyl-1,2,3-butanetricarboxylic acid (MBTCA). Monomeric and dimeric reaction products were accurately identified in the mass spectra, even at a total OA mass concentration < 2 µg m$^{-3}$. The present study showed that the HEAC/ESI-Orbitrap-MS system is a robust technique for the real-time chemical characterisation of OA particles under atmospheric relevant conditions.

# 1 Introduction

Aerosols are small solid or liquid particles suspended in the atmosphere. They can be categorised by aerodynamic diameter into coarse ($d_a$ > 2.5 µm), fine (0.1 µm < $d_a$ < 2.5 µm) and ultrafine ($d_a$ < 0.1 µm) particles.(Kulkarni et al., 2011) Chemically, aerosols rich in inorganic materials like sulphates and sea salt are called inorganic aerosols, while those dominated by organic matter are called organic aerosols (OA). Previous studies have shown that OA constitutes 20-90% of the fine particle mass in tropospheric aerosols (Kanakidou et al., 2005; Jimenez et al., 2009). Aerosols significantly affect air quality and human health



(Pöschl, 2005; Huang et al., 2014), leading to extensive studies on their physical and chemical properties (Tao et al., 2017; Quinn et al., 2015; Fan et al., 2016; Shrivastava et al., 2017). Fine and ultrafine aerosols are particularly important due to their complex environmental reactions and deep penetration into the respiratory system (Shiraiwa et al., 2017; Kwon et al., 2020; Bates et al., 2019). Aerosols also impact global climate by directly affecting the radiative budget through the absorption and reflection of radiation and indirectly by acting as cloud condensation nuclei (CCN) (Mahowald et al., 2011; Mcneill, 2017).

Changes in aerosol chemical composition can alter their optical properties and CCN formation abilities, affecting their radiative forcing. Therefore, understanding aerosol chemical composition is crucial for assessing their environmental, health, and climate impacts.

Traditionally, molecular-level characterisations of aerosols were typically done through offline analyses. Aerosol samples were collected on filters, followed by sample extractions, derivatisation, and instrumental analyses (Fleming et al., 2020; Sun

et al., 2016). The most utilised techniques for identifying and quantifying chemical species in aerosols are Gas Chromatography/Mass Spectrometry (GC/MS) and Liquid Chromatography/Mass Spectrometry (LC/MS) (Kautzman et al., 2010; Budisulistiorini et al., 2015). Offline analyses are advantageous due to well-developed protocols and extensive databases, facilitating the chemical characterisation of aerosol samples. Various techniques can be used to obtain detailed information on the chemical compounds under study. For example, ion mobility spectrometry–mass spectrometry (IMS-MS)

can be used to differentiate between isomers of compounds with the same chemical formula (Krechmer et al., 2016). However, they lack time resolution, making it challenging to track changes in chemical composition over time. Additionally, offline methods require several sample preparation steps, which can introduce errors and alter the sample composition (Miljevic et al., 2014).

As the evolution of the aerosol's chemical composition is critical for understanding its chemistry in different processes, online

techniques have been developed to tackle this challenge. The Aerosol Mass Spectrometer (AMS) is widely used for real-time bulk chemical composition analysis, providing crucial information on the chemical and physical properties of aerosols. However, AMS uses a hard ionisation source (electron impact ionisation), resulting in mass spectra with mostly fragment ion peaks, which are grouped into organics and several inorganic ions such as sulphate, nitrate, chloride, and ammonium (Canagaratna et al., 2007). Recently, CHARON (CHemical analysis of AeRosol ONline) inlet coupled with a proton transfer

reaction time-of-flight mass spectrometer (PTR-TOF-MS) and extractive electrospray ionisation- (EESI) TOF-MS have been developed for the real-time, direct molecular-level analysis of aerosol chemical composition in ambient air (Lopez-Hilfiker et al., 2019; Piel et al., 2019). However, these instruments have drawbacks. The PTR-TOF-MS can only detect compounds with proton affinities higher than that of $H_2O$, and switching ion sources to detect a wider array of compounds may require changes in instrumental configurations. Additionally, the TOF mass analyser used in both instruments cannot unambiguously separate

isobaric compounds, making post-experiment data analysis challenging.

Several studies have employed a different approach to obtain a time-resolved chemical composition profile of fine particles using a particle-into-liquid sampler (PILS) coupled with various mass spectrometry systems for online chemical characterisation of aerosols. For example, Zhang et al. (2016) coupled PILS with Ultra Performance Liquid



Chromatography/ESI Quadrupole Time-of-Flight/Mass Spectrometer (UPLC/ESI-Q-TOFMS) to analyse atmospheric aerosol

samples, demonstrating sufficient sensitivity and detection limits for single compounds at microgram to nanogram per cubic

meter concentrations. Their system was used to chemically characterise the reactive uptake products of isoprene epoxydiols

on acidic ammonium sulphate aerosols. Saarnio et al. (2013) combined PILS with high-performance anion-exchange

chromatography (HPAEC)-MS to measure levoglucosan in ambient aerosols, while Parshintsev et al. (2010) used PILS with

solid-phase-extraction (SPE)-LC/ion-trap MS to investigate organic acids in aerosols. While the separation techniques

employed by the above studies benefit the chemical characterisation of the OA samples, the downside is that they are only

partially real-time. This is because PILS-collected samples were either stored in vials or the LC system's sample injection loop

before MS analysis, with time resolution depending on the collection or injection frequency. For example, the time resolutions

for PILS-UPLC/ESI-Q-TOFMS and PILS-HPAEC-MS were five and eight minutes, respectively. Additionally, improving the

mass resolution of these MS systems could enhance peak separation and identification of reaction products. Therefore, the aim

of the current study was to construct a new system that provides real-time determination of aerosol chemical composition at

atmospheric-relevant mass loadings. To achieve this, we used a custom-built aerosol collection system described in Brown et

al. (2019b) and coupled it to a high-mass resolution mass spectrometer, namely Orbitrap MS.

This high-efficiency aerosol collector (HEAC) achieves nearly 100% collection efficiency for fine and ultrafine particles down

to 30 nm. The HEAC operates similarly to the particle-into-liquid sampler (PILS), using condensational growth of aerosol

particles to enhance their collection efficiency. While PILS uses a washed-wall collector, HEAC employs a vortex collector—

a cyclone with a standing liquid vortex on its wall during operation (Orsini et al., 2008).

In recent years, several high-resolution mass spectrometric techniques have been commercialised, allowing researchers to

obtain detailed molecular information with minimal peak overlap. One such technique is the Orbitrap mass analyser coupled

with an ESI source (Zubarev and Makarov, 2013). The ESI-Orbitrap-MS offers a mass resolution above 100,000, suitable for

analysing complex mixtures containing numerous compounds, such as organic aerosol (OA) samples. Using an ESI source

can also preserve the molecular identity of chemical compounds in the sample.

The proposed system combines the HEAC for aerosol collection with the ESI-Orbitrap-MS for chemical characterisation. The

system's sensitivity and detection limits were assessed using various chemical compounds, and its performance was evaluated

by analysing the products of α-pinene ozonolysis.

**2 Methodology**

**2.1 Coupling of HEAC and ESI-Orbitrap-MS**

Detailed design and characterisation of the HEAC can be found elsewhere (Brown et al., 2019b). Briefly, the aerosol sample

flows through the HEAC inlet at a typical rate of 16.7 L min$^{-1}$, reaching a condensation growth chamber. Here, the aerosol

encounters a parallel jet of steam, and the rapid cooling of the steam creates supersaturated conditions in the chamber. This

causes the aerosol particles to grow into droplets a few microns in size. These droplets were collected by the conical vortex





collector at the chamber exit. When in operation, a vortex working fluid flows from the top to the bottom of the vortex collector and forms a thin film of liquid on the walls of the cyclone. Detailed operation principles of the vortex collector are available elsewhere (Brown et al., 2019b; Orsini et al., 2008). The working fluid flow rate out of the vortex collector was set to 1 ml min$^{-1}$ during method development, consistent with previous applications (Brown et al., 2020; Brown et al., 2019a). The collected aerosol sample can be analysed instrumentally without further preparation.

The ESI-Orbitrap-MS (LTQ Orbitrap Elite, Thermo Fisher, USA) in this study is managed by the Central Analytical Research Facility (CARF) at Queensland University of Technology (QUT). To connect the HEAC to the ESI source of the MS, the vortex collector outlet can be linked directly to the ESI sample inlet. However, the peristaltic pump (ISMATEC Reglo ICC 4 channel, Cole-Parmer, USA) controlling the HEAC's liquid flows does not generate enough pressure to deliver the sample through the ESI source's capillary nozzle. To address this, a high-pressure auxiliary pump (Dionex AXP, Thermo Fisher, USA) was used to draw some liquid from the vortex collector and inject it into the ESI source. Figure 1 shows a schematic diagram of the setup. The sample flow rate to the ESI source was set to 0.5 ml min$^{-1}$.

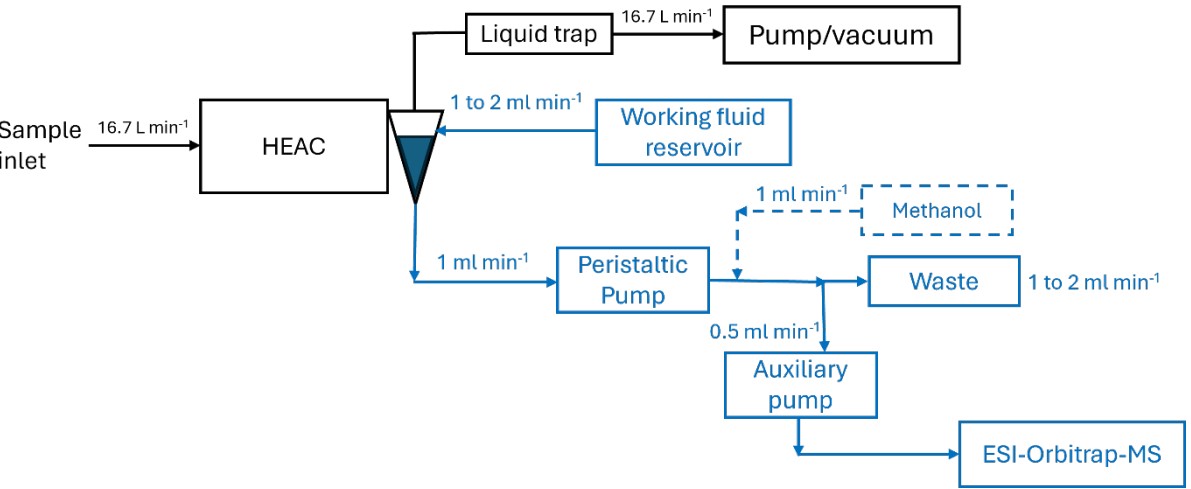

**Figure 1. Schematic diagram of the HEAC/ESI-Orbitrap-MS system. Components in blue indicate liquid flow paths and flow rates. Dashed lines indicate the additional setup when using Milli-Q water as the working fluid.**

### 2.2 Characterisation of the HEAC/ESI/Orbitrap-MS system

The characterisation of the HEAC/ESI/Orbitrap-MS system is divided into two parts. The first part evaluates the system's sensitivity and detection limits for various chemical compounds under different conditions. The second part assesses the system's ability to analyse a complex organic aerosol sample generated from α-pinene ozonolysis.

### 2.2.1 Sensitivity and detection limit of the HEAC/ESI-Orbitrap-MS system

To characterise the sensitivities and detection limits of the HEAC/ESI/Orbitrap-MS system, the mass concentration of a chemical compound measured before entering the HEAC is compared with the compound's signal intensity measured by the





ESI-Orbitrap-MS. Figure S1 in Supporting Information (SI) shows a schematic of the setup for single compound characterisation. Briefly, each examined compound was nebulised by an atomiser (Collison 1-jet, CH Technologies, USA) and then pushed through a diffusion dryer filled with silica gel. The sample flow from the dryer was mixed with filtered air to provide a sufficient flow volume for the HEAC (16.7 L min$^{-1}$). Before entering the HEAC, the mass concentration of the diluted sample was measured by a Scanning Mobility Particle Sizer (SMPS) system (electrostatic classifier 3080 and condensation particle counter 3776, TSI Incorporated, USA, with a custom-built column).

To estimate the sensitivity of a particular chemical compound in the HEAC/ESI/Orbitrap-MS system, a calibration curve of mass concentration versus ESI/Orbitrap-MS signal is needed. The mass concentration entering the system was varied by adjusting the compressed air flow rate into the atomiser. For each concentration, the ESI-Orbitrap-MS signal was measured continuously for at least 5 minutes to obtain a steady average. The sample mass concentrations measured by the SMPS were also averaged. Sensitivity is determined by finding the slope of the best-fit line of the calibration curve. Details of the calculation can be found in SI.

Six chemical compounds—vanillic acid (VA), adonitol, erythritol, tricarballylic acid (TCA), sucrose, and trehalose—were used for sensitivity and LOD experiments due to their varying water solubilities, volatilities, and relevance to atmospheric chemistry. Their physical properties are detailed in Table S1 in the SI. Milli-Q water was used as the working fluid for the sensitivity and LOD characterisation experiments. To assist the ionisation of samples, methanol was added to the sample flow via a T-junction before the auxiliary pump whenever Milli-Q water was used as the working fluid.

To account for variations in signal intensity from the ESI process, malic acid (~0.1 µM) was added to the vortex collector's working fluid. All ESI/Orbitrap-MS data in the sensitivity and LOD calculations were corrected based on the malic acid signal. The study also examined the sensitivity and LOD of the HEAC/ESI-Orbitrap-MS using different working fluids, as the ionisation efficiency and, thus, sensitivity and LOD can be significantly affected by the choice of solvent. The solvents tested were Milli-Q (MQ) water, methanol, and acetonitrile (ACN), which are commonly used in ESI-MS. The experimental procedures remained consistent with the sensitivity and LOD characterisation experiments, except for the variation in the working fluid.

**2.2.2 α-pinene ozonolysis experiment**

To validate the HEAC/ESI-Orbitrap-MS system, α-pinene ozonolysis was conducted in a custom-built 5 L reaction bottle, and the particle-phase products were analysed by the system. Two types of α-pinene ozonolysis experiments were performed: fast injection and slow injection. The setups for both are shown in Figure S1 in SI. In both experiments, α-pinene ozonolysis occurred in the reaction bottle. Ozone was generated by pushing 0.3 L min$^{-1}$ compressed air through an ozone generator (SOG-2, Analytik Jena US, USA), achieving a concentration of ~2 ppm in the bottle. For the fast injection, 20 µL of α-pinene was directly injected into the reaction bottle through a small hole in the cap, evaporating in less than five minutes. For the slow injection, 100 µL of α-pinene in a three-necked round bottom flask was placed before the reaction bottle, with ozone passing through it first. This setup slowed the evaporation rate of α-pinene, taking 15-20 minutes for complete evaporation.





After passing through the reaction bottle, the product mixture was pushed through a diffusional dryer containing charcoal (hereafter referred to as a charcoal filter) to remove the excess ozone and gas-phase products. Afterwards, the gas stream was mixed with a makeup flow of charcoal- and HEPA-filtered lab air to compensate for the sampling volume of the HEAC. For the fast α-pinene injection experiment, the mixed sample flow was sampled by the custom-built SMPS for its size distribution
before being analysed by the HEAC/ESI-Orbitrap-MS system. The mass concentration of particles was calculated from the measured size distribution assuming a particle density of 1.2 g cm$^{-3}$ (Shilling et al., 2009). For the slow α-pinene injection experiment, after the sample was mixed with the make-up flow, the sample flow went through one of the three scenarios before going into HEAC: (1) passing through a HEPA filter, (2) passing through a charcoal filter, and (3) passing through no filter. This sequence was repeated four times, and the switching between each scenario was carried out manually by a three-way
valve. The rotation of different scenarios aims to examine the system's response time and to determine whether the samples are all in the particle phase.

It should be noted that the α-pinene ozonolysis experiment carried out in the current study serves to prove the functionalities of the new HEAC/ESI-Orbitrap-MS system. Atmospheric relevance was not considered in the experimental design, and the results should not be used directly to compare with other studies under atmospheric-relevant conditions.

**2.3 Product identification in α-pinene ozonolysis experiments**

A third-party software, MZmine (ver. 3.9) (Schmid et al., 2023), was used for product identification in α-pinene ozonolysis experiments. Detailed workflow and parameterisation of the software were outlined in the SI. Briefly, the experiment's raw data file contains a time profile of mass spectra corresponding to the real-time measurement of samples collected by the HEAC at a time resolution of around 0.8 seconds. After the raw data was imported to Mzmine, "Mass detection" was done to identify
all the ions present in every mass spectrum with signal intensities above certain thresholds (e.g. noise level and separations between ion signals). A mass list was generated for all mass spectra. A "feature detection" algorithm was then applied to the raw data to create the time series of all ions in the mass list that fulfil the criteria set in the algorithm (Pluskal et al., 2010; Myers et al., 2017). At the current threshold settings, 204000 ions of different m/z were identified by the "mass detection" algorithm, and the time series of 20004 ions were obtained from the "feature detection" function. These time series were then
individually inspected. Ions were assigned as "product ions" if their signals increased when the HEAC was sampling from the reaction bottle.





## 3. Results and discussion

### 3.1 Characterisation of the HEAC/ESI/Orbitrap-MS system

#### 3.1.1 Sensitivity and LOD

Nebulisation of aqueous solutions of the six chemical standards, namely adonitol, erythritol, sucrose, trehalose, VA and TCA, resulted in aerosol samples with mass concentrations ranging from 0.1 to 30 µg m$^{-3}$. Calibration curves of the chemical standards are illustrated in Figure 2. Generally, the ESI-Orbitrap-MS signal intensities of the standards increase linearly with their particle mass concentration. Their sensitivities and LODs are summarised in Table S2. The sensitivities of the standards tested in the present study ranged from $0.53 \pm 0.11 \times 10^4$ (trehalose) to $1.9 \pm 0.21 \times 10^4$ (TCA) counts m$^3$ µg$^{-1}$. The chemical

standards used can be divided into three categories: organic acids (TCA and VA), sugars (sucrose, trehalose) and polyols (adonitol and erythritol). Among these three categories, organic acids have the highest sensitivities. The sensitivities of sugars and polyols fall below $0.7 \times 10^4$ counts m$^3$ µg$^{-1}$. A possible explanation for the above observation is their difference in ionisation efficiencies under negative ESI mode (Oss et al., 2010). In general, compounds with a stable anionic form, such as acids, will be efficiently ionised by ESI in the negative ion mode (Kamel et al., 1999; Ehrmann et al., 2008). The anions of sugars and

polyols are less stable, and their ionisation efficiencies are lower than those of organic acids. Therefore, the sensitivities of organic acids are higher than those of sugars and polyols in the present study. The above results show that the functional group of the compound strongly affects its sensitivity in the HEAC/ESI-Orbitrap-MS system.

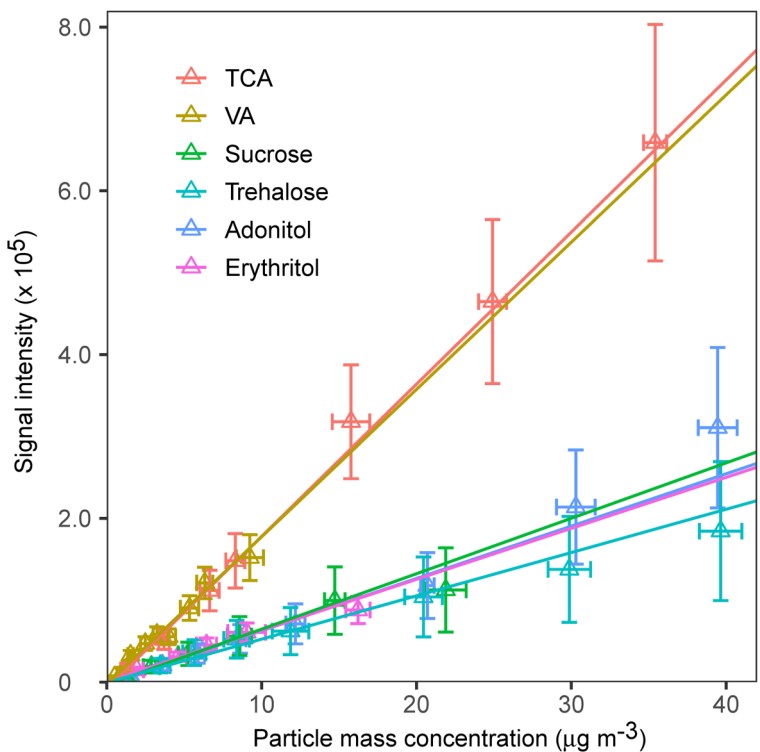



**Figure 2. Calibration curves of the six chemical compounds used for sensitivity and LOD determination.**


The chemical standard's LOD was calculated from its sensitivity and background signal's standard deviation, as described in SI. Among the chemicals tested in the present study, erythritol has the lowest LOD of $1.1 \pm 0.14$ ng m$^{-3}$. On the other hand, TCA has the highest LOD of $65 \pm 7.4$ ng m$^{-3}$. The LOD result has the same order of magnitude as those reported in the literature using similar instrumental techniques. Lopez-Hilfiker et al. (2019) determined the LOD of raffinose and dipentaerythritol of

their EESI-TOF instrument to be a few ng m$^{-3}$. Zhang et al. (2016) reported the LOD of a wide range of chemical standards, including organic acids and polyols, to be tens of ng m$^{-3}$ to a few µg m$^{-3}$ in their particle-into-liquid-sampler (PILS) coupled with UPLC/ESI-Q-TOFMS system. The slightly lower LOD reported in the present study is probably the result of using Orbitrap-MS as the analyser, which provides a higher mass resolution and, thus, a lower background noise signal than a TOF-MS.

**3.1.2 The effect of working fluid on the sensitivity and LOD**

To investigate the effect of different solvents as the working fluid in the vortex collector, the sensitivities and LOD of vanillic acid, TCA, and adonitol were determined using Milli-Q water, methanol, and ACN as the working fluids. The results are summarised in Table S3.

In general, the choice of working fluid did not significantly affect the LOD of the chemical compounds, as LOD is primarily

influenced by the standard deviation of the compound's background signal. However, significant differences in sensitivity were observed among the three working fluids. For the two acids, using ACN as the working fluid resulted in more than twice the sensitivity compared to using methanol or Milli-Q water. For adonitol, Milli-Q water yielded the highest sensitivity, while methanol resulted in the lowest sensitivity for TCA and VA.

Unlike LOD, many factors influence the signal intensity and sensitivity of a compound in ESI-MS, such as the ionisation

efficiency during the ESI process, the ionisation energy of the compound in the solvent, and the ion suppression effect caused by other ions in the sample. These factors have mixed effects on the signal intensity and sensitivity of the target compound. Results from this study show no clear pattern for the three working fluids and compounds used. Further research is needed to understand the relationship between the working fluid in the vortex collector and the compound's sensitivity measured by the ESI-Orbitrap-MS, especially when quantitative analysis is required.

**3.1.3 Effects of ESI heater temperature on the signal intensity**

To demonstrate the effects of ESI settings on the target compound's signal intensity, the heater temperature of the ESI source was altered while the target compound's concentration introduced to the HEAC/ESI-Orbitrap-MS system was kept constant. The reason for choosing to vary the heater temperature is that it controls the temperature of the auxiliary nitrogen gas, which is used to assist the desolvation of sample solutions. Therefore, it is anticipated that the heater temperature will significantly





affect the evaporation of the charged solvent droplet and the ionisation of the target analyte. In this experiment, erythritol was nebulised at a fixed concentration and measured by the HEAC/ESI-Orbitrap-MS system. The ESI heater temperature was increased from 50 to 350 °C with a 50 °C increment. Figure 3 shows the signal intensity of erythritol and the total ion count (TIC, from m/z = 50 to 500) in response to the change in heater temperature. It can be observed in the figure that the TIC doubles for every 100°C increase in heater temperature. This observation agrees with the fact that the rise in heater temperature

will enhance the ionisation efficiency of the sample by promoting solvent evaporation in the vaporisation chamber of the ESI source. Similarly, the signal intensity of erythritol increases with heater temperature from 50 to 250 °C. However, when the heater temperature rises above 250 °C, the signal intensity of erythritol starts to drop. This observation likely resulted from the ion suppression effect on the erythritol signal by the high background signal, as reflected by the increase in TIC (Furey et al., 2013). The suppression effect was also observed in background ions (e.g. m/z = 107 and 109) unrelated to erythritol. Another

possible reason for the decrease in the ion signal of erythritol at high heater temperature is the thermal decomposition of the compound. However, since there is no reported thermal decomposition temperature of erythritol in the literature, the exact reason for the observed decrease in its signal intensity under high temperatures cannot be concluded in the current study. Nevertheless, the current set of experiments showed that ESI settings, such as heater temperature, will significantly affect the signal intensity of the target compound, and the selection of ESI parameters should be considered during the design of the

experiment, especially if quantitative analysis is desired.

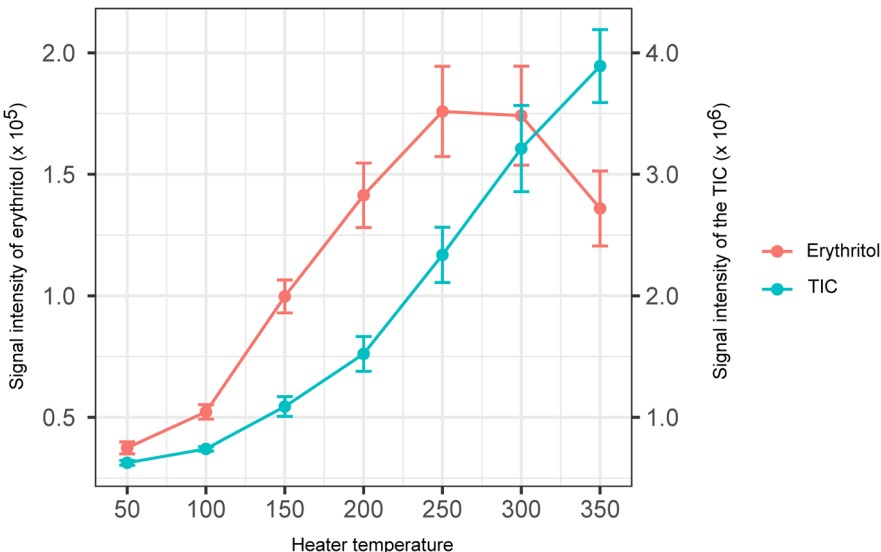

**Figure 3. The signal intensity of erythritol and the total ion count (TIC, from m/z = 50 to 500) under different heater temperatures.**



### 3.2 α-pinene ozonolysis experiments

#### 3.2.1 Fast α-pinene injection experiment

Figure 4 shows the mass concentration of particles after injecting 20 µL of α-pinene into the reaction bottle. It can be observed that particles formed almost immediately after the α-pinene injection, and the mass concentration peaked around five minutes after the start of the experiment. After that, the particle mass concentration decreased gradually until reaching the background level. The peak mass concentration was around 30 µg m$^{-3}$ in the fast α-pinene injection experiment.

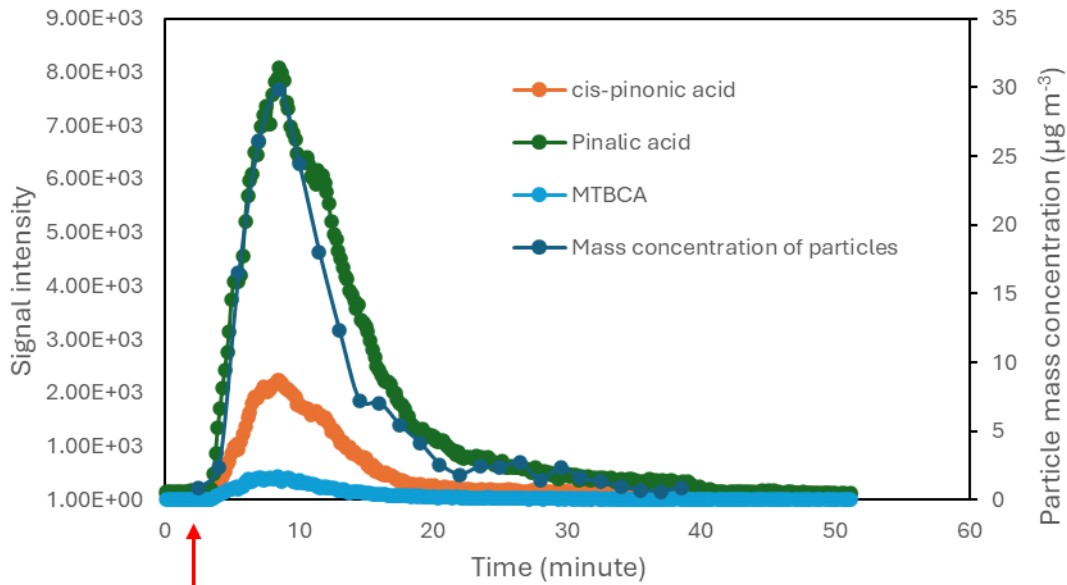

**Figure 4. The mass concentration of particles measured by the SMPS and the signal intensity of selected α-pinene ozonolysis products captured by the ESI-Orbitrap-MS for the fast α-pinene ozonolysis experiment. The red arrow marks the injection time of α-pinene.**

A typical mass spectrum of the α-pinene ozonolysis products is shown in Figure 5. Using the product identification workflow described in the methodology section, over three hundred ions were assigned as products in the fast α-pinene injection experiment, including isotope signals of some product ions. Table S4 summarises the product ions observed in the experiment

(excluding the isotopes) with a chemical formula predicted from the exact mass of each ion. Figure 6 shows the mass defect plot of α-pinene ozonolysis products identified in the present study. These products span a wide range of carbon numbers, ranging from two to 26 carbon atoms. The bottom left cluster of products in Figure 6 corresponds to monomeric compounds with small carbon numbers and low mass defects. Apart from monomeric compounds, dimeric products were observed in the fast α-pinene ozonolysis experiment, denoted by the upper right cluster of ions in Figure 6. This pattern is similar to those

observed by Zhang et al. (2017) and Wang et al. (2021), which utilised offline LC-MS techniques for the chemical analysis of the α-pinene ozonolysis products. The similarity between the α-pinene ozonolysis products observed in the current study and





the literature emphasises the robustness of the HEAC/ESI-Orbitrap-MS to analyse the chemical composition of organic aerosols.

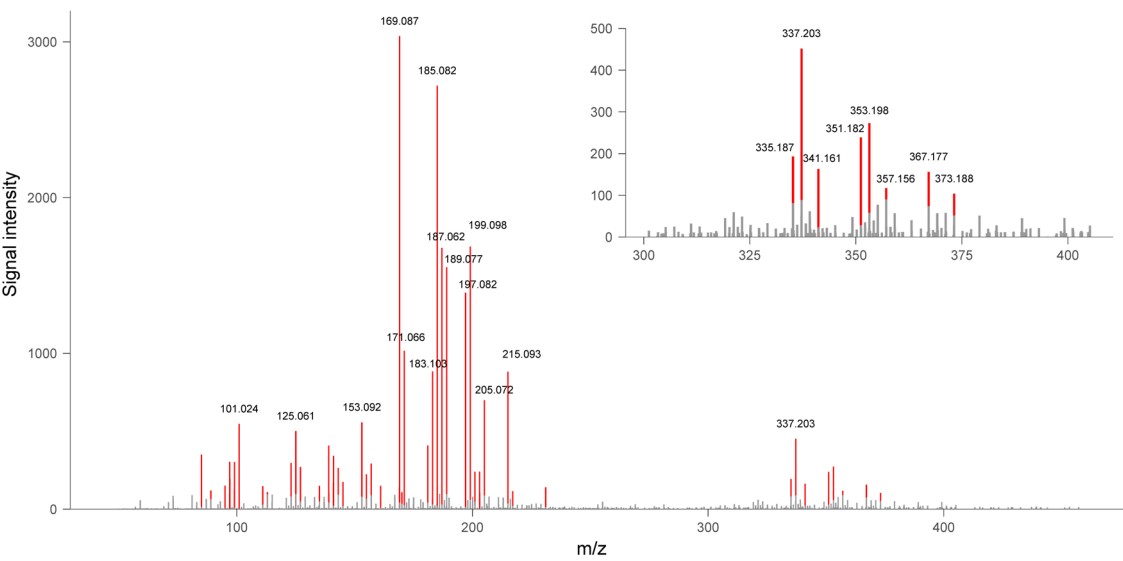

**Figure 5.** The mass spectrum obtained from the fast α-pinene injection experiment when the particle mass concentration was the highest. The mass spectrum is background subtracted. Peaks with signal intensity larger than 100 are highlighted in red, and the m/z for peaks with signal intensity larger than 450 are labelled in the figure. The upper right panel shows the magnification of the m/z = 300 to 400 range of the same mass spectrum. Peaks with signal intensity larger than 100 are highlighted in red and labelled with their corresponding m/z.

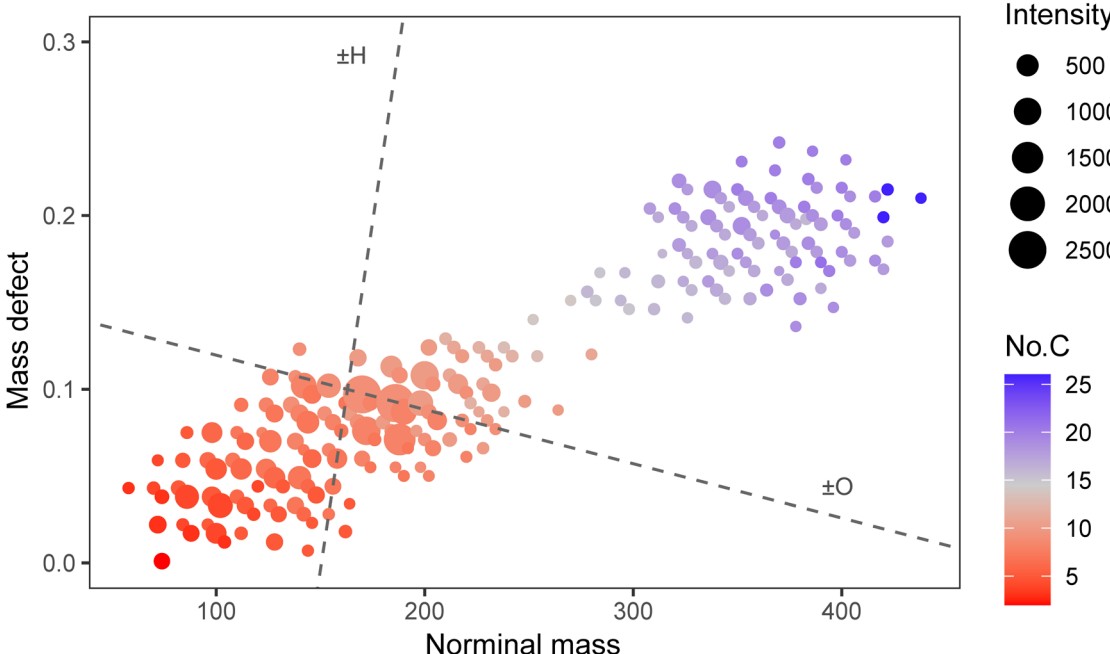





**Figure 6. The mass defect plot of the ozonolysis products in the fast α-pinene injection experiment. The data in the figure correspond to the moment when the particle mass concentration peaked.**

By comparing the exact mass of the product ions with α-pinene ozonolysis products in the literature, some of the ions were assigned chemical structures. These compounds are summarised in Table S5. Figure 4 shows the time series of selected α-pinene ozonolysis products, including cis-pinonic acid, pinalic acid, and 3-methyl-1,2,3-butanetricarboxylic acid (MBTCA). As shown in the figure, the signal intensities of these products follow the particle mass concentration trend measured by the SMPS. Indeed, all the identified products showed the same signal intensity trend as in Figure 4. This is probably due to the short reaction time and high initial α-pinene concentration, which suppressed the formation of higher-generation products in the present experiment. Figure S2 shows the hydrogen-to-carbon (H/C) and oxygen-to-carbon (O/C) ratios of the identified products. The figure shows that most products have an O/C ratio smaller than ~0.6, indicating a lower degree of oxidation (Molteni et al., 2019). Nevertheless, the agreement between the particle's mass concentration and the products' signal intensities confirmed the capability of the HEAC/ESI-Orbitrap-MS system to identify the chemical composition of organic aerosols in real time.

### 3.2.2 Slow α-pinene injection experiment

In the slow α-pinene injection experiment, the evaporation rate of α-pinene was slowed down, and a filter-rotation setup was added before the HEAC, as described in the methodology section. Figure 7 summarises the mass concentration of particles generated from the slow α-pinene injection experiment. As shown in the figure, a rapid increase in particle mass concentration is observed at the beginning of the first rotation cycle, which marks the start of the α-pinene ozonolysis reaction. In all four rotation cycles, the particle's mass concentration decreases when the HEAC inlet is switched to the charcoal filter and drops to almost zero when changed to the HEPA filter. The rotation between filters sometimes causes a sharp increase in the particle mass concentration because the manual switching of the three-way valve causes pressure fluctuation. The mass concentration of particles decreases during the charcoal filter phase due to the particle loss in the filter. Figure 7 also shows the changes in signal intensities of selected reaction products measured by the ESI-Orbitrap-MS. It can be observed that the products' signal intensities followed the trend of particle mass concentration. The HEAC/ESI-Orbitrap-MS system's response time, which is the time required for the ESI-Orbitrap-MS signal to reach a steady state after changes in sample concentration, can be estimated from the product's signal intensity time series. As shown in Figure 7, the increase and decrease in the product's signal intensity take around two minutes to stabilise after each filter change. The response time is mainly governed by the length of the tubing connecting the vortex collector's outlet and the ESI source's inlet. In other words, minimising the length or decreasing the inner diameter of the tubing will shorten the system's response time, which is beneficial in applications involving fast sample composition changes.





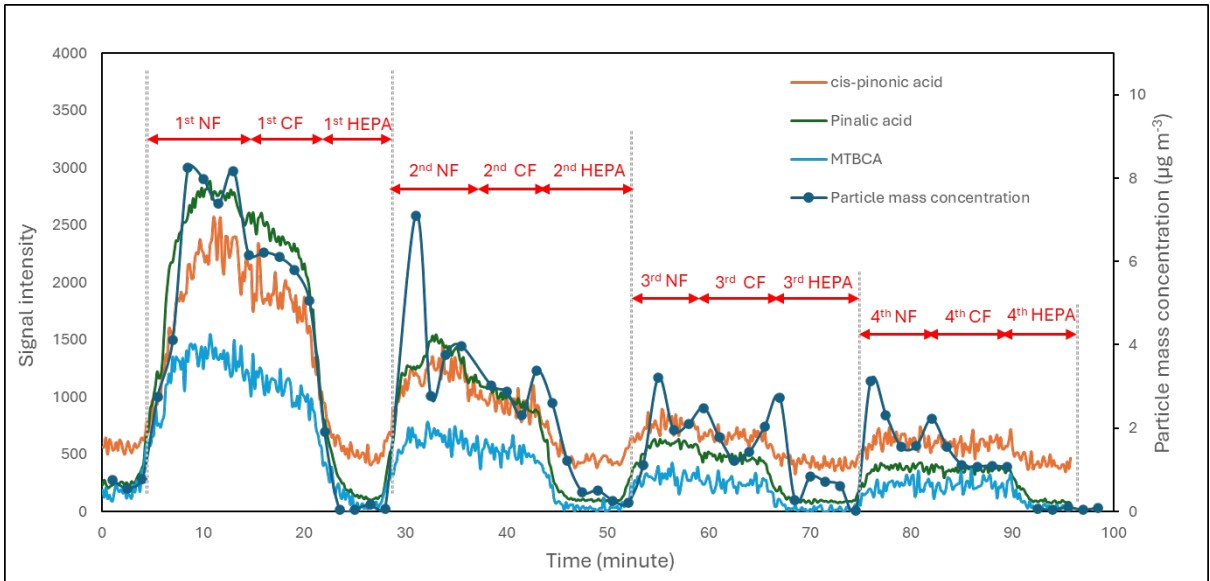

**Figure 7. Mass concentration of particles generated from the slow α-pinene injection experiment and the signal intensity of selected reaction products measured by the ESI-Orbitrap-MS. For each filter rotation cycle, the sample flow went through one of the three scenarios before going into HEAC: (1) passing through a HEPA filter (HEPA), (2) passing through a charcoal filter (CF), and (3) passing through no filter (NF).**

A typical mass spectrum of the products in a slow α-pinene injection experiment during different phases is shown in Figure S3. The mass spectrum in the no-filter and charcoal filter phases is similar to that obtained from the fast α-pinene injection experiment. This is because, in both experiments, a charcoal filter was connected downstream to the reaction bottle to remove the excess ozone and gas phase species in the reaction. Samples reaching the HEAC are thus only in the particle phase. This is confirmed by the mass spectrum in the HEPA filter phase, of which the product signals all dropped to background levels. Another finding in the slow α-pinene injection experiment is that the HEAC/ESI-Orbitrap-MS can analyse samples with low mass concentration. As outlined in Figure 7, the mass concentration of α-pinene ozonolysis product decreased over time. During the no-filter phase in the fourth rotation cycle, the mass concentration of particles dropped to below two μg m$^{-3}$. This particle mass concentration is similar to those observed in clean environments. Figure 8 shows the mass spectrum of products during the no-filter phase in the fourth rotation cycle and the time profile of a dimeric reaction product. Despite the low sample concentration, most reaction products are still measurable. The time profile of the dimer showed an apparent increase in signal intensity during the fourth no-filter phase. This observation emphasised the ability of the HEAC/ESI-Orbitrap-MS system to analyse the chemical composition of organic aerosols in atmospheric-relevant conditions.




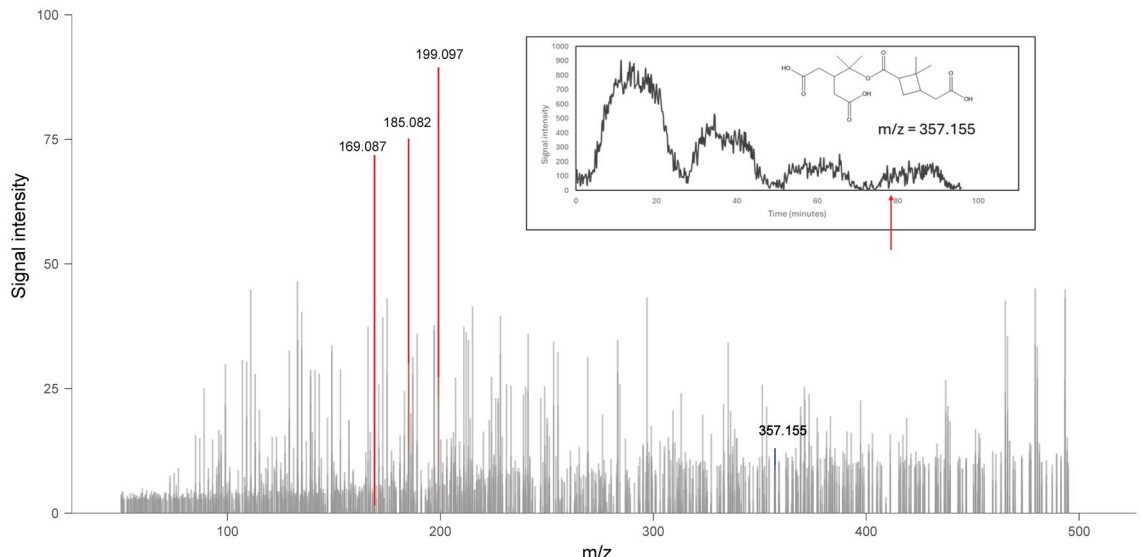

**Figure 8. Mass spectrum of products during the no-filter phase in the fourth rotation cycle in slow α-pinene injection experiment. Peaks with signal intensity larger than 50 were highlighted in red. The plot embedded in the mass spectrum corresponds to the time profile of a dimeric product with m/z = 357.155. The red arrow in the plot corresponds to the time when the mass spectrum is obtained.**

## 4. Limitations and practical considerations

The results of the current study showed that the proposed HEAC/ESI-Orbitrap-MS system has high sensitivities and low LODs toward chemical standards and can analyse complex OA samples in real-time. Nevertheless, limitations exist for the HEAC/ESI-Orbitrap-MS system, which will be discussed together with practical considerations when using the system in the following sections.

### 4.1 The choice of working fluid

The HEAC/ESI-Orbitrap-MS system can only identify the chemical composition of the OA fraction soluble in the working fluid because ESI cannot efficiently ionise undissolved compounds. This is a common limitation in all analytical systems utilising ESI to generate sample ions. One trivial way to tackle this problem is to change the working fluid to another solvent in which the target compounds have higher solubility. However, as shown in our results, changing the working fluid will significantly impact the signal intensity and sensitivity of the target compound. Furthermore, although the vortex collector can accommodate most organic solvents, not all are compatible with the ESI-MS. Therefore, it is crucial to identify the compounds of interest during the experimental planning stage and consider which solvent to use as the working fluid. If possible, preliminary tests should be done to test the performance of a few different candidates. If there are no specific target compounds, using methanol or ACN as the working fluid is recommended to cover a solubility range as wide as possible.





### 4.2 The response time of the system

Although the HEAC/ESI-Orbitrap-MS is capable of continuous sampling and analysis of OA samples, the time resolution of the instrument depends on the response time of the system, which is the time required for a change in sample composition in front of the HEAC to be reflected by the mass spectrum. As shown in the slow α-pinene injection experiment, it takes around two minutes for the compound signals to reach a steady state whenever there is a sudden change (e.g. switching the HEAC inlet from no filter to HEPA filter) in the sample concentration. Therefore, it can be considered that the compound's signal measured by the MS represents a two-minute-average concentration of the compound in the OA sample. This, in turn, means that any changes in sample composition and concentration within the two-minute time frame will not be observable by the MS. Although such rapid changes are rare in atmospheric chemistry, it is desirable to shorten the response time and increase the time resolution of the system as much as possible. A straightforward way to do that is to shorten the tubing connecting the outlet of the vortex collector and the inlet of the ESI source. Replacing the piston pump in the current setup with a liquid handling system of smaller internal volume can also minimise the time required for the liquid sample to reach the ESI source.

### 4.3 Ion suppression and matrix effects of ESI

Ion suppression and matrix effects are well-known issues with ESI sources, hindering their application in accurate quantitative analyses of complex samples like organic aerosols (Parshintsev and Hyötyläinen, 2015). Separation techniques, such as LC and SPE, can be applied before ESI-MS analyses to reduce the influence of ion suppression and matrix effects by reducing the amount of co-eluting compounds. However, the additional separation step will reduce the time resolution of the whole system. Since the objective of the present study is to develop a real-time system for the chemical characterisation of organic aerosol with a high time resolution, samples collected by the HEAC were introduced to the ESI source in a direct infusion manner. To minimise ion suppressions in the current setting, the total concentration of analytes should be kept at a low level ($< 10^{-5}$M) (Furey et al., 2013). This level corresponds to a mass concentration of around 100 µg m$^{-3}$ of soluble chemical components in the aerosol sample, assuming an average molecular mass of 200 for the soluble compounds. An internal standard can also be used to monitor the ion suppression effect during the sampling period. The present study added malic acid to the working fluid as an internal standard in fast and slow α-pinene injection experiments. The malic acid signal did not change in both cases as the α-pinene ozonolysis products reached the MS, as shown in Figure S4. The above observation showed that ion suppression was minimal at the current experimental conditions and OA mass loading. However, in experiments with high OA mass loadings (e.g. over 200 µg m$^{-3}$), attention must be paid to the ion suppression and matrix effects if doing quantitative data analyses.

### 4.4 Other considerations in experimental designs

As described by Brown et al. (2019b) the HEAC collects samples into the working fluid regardless of their solubility. A high insoluble material content in the sample flow might block the ESI nozzle and decrease the spray's performance. In experiments

anticipated to have a high emission rate of insoluble particles, such as biomass burning simulations, inline filters should be used to remove the insoluble particles before the ESI source. Besides, the current prototype HEAC/ESI-Orbitrap-MS setup utilised a single piston pump for the sample injection. Although able to achieve continuous sample injection, the single-piston

pump also introduced significant pulsations to the sample flow. Replacing the pump with a double piston pump or other liquid handling system with minimal pulsation can potentially increase the performance of the system and the readability of the mass spectrum.

## 5. Conclusion

The current study introduces an innovative system for the real-time chemical characterisation of organic aerosol particles. The

system combines a HEAC for sample collection with an ESI-Orbitrap-MS for chemical analysis. The synergy of these instruments enables continuous sampling and real-time chemical analysis. Results demonstrate that the HEAC/ESI-Orbitrap-MS system offers comparable or even better sensitivity and LOD compared to similar techniques. α-pinene ozonolysis experiments further confirm the system's ability to provide real-time chemical insights of a complex organic aerosol sample. With its soft ionisation source, the system identifies intact chemical compounds, minimising molecular fragmentation. The

Orbitrap-MS delivers high-mass-resolution spectra, facilitating precise product identification and isobaric ion separation. Even under low OA mass concentrations ($< 2\mu g\ m^{-3}$), the system accurately identifies OA composition, highlighting its ability to chemically characterise OA at atmospheric-relevant mass loadings.

## Author contributions

YL and BM designed the experiment. YL carried out the experiment, analysed the data, and wrote the manuscript. BM and

ZR supervised and provided comments on the experimental work. All the authors reviewed and provided feedback on the manuscript.

## Competing interests

The authors declare that they have no conflict of interest.

## Acknowledgements

This work was enabled by the use of the Central Analytical Research Facility (CARF) at the Queensland University of Technology (QUT). The authors thank Dr David Marshall from CARF for his assistance with the ESI-Orbitrap-MS setup and data analysis.



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
