# Peer review of "The Coupling of a High-efficiency Aerosol Collector with Electrospray Ionisation/Orbitrap Mass Spectrometry as a novel tool for Real-time Chemical Characterisation of Fine and Ultrafine Particles"

_Atmospheric Measurement Techniques, 2024_

## Author Response (AR1)

**Response to reviewer 1**

This paper describes a measurement system that couples two sophisticated analytical techniques; a vortex wet chemical aerosol collector, and electrospray Orbitrap high resolution mass spectrometry (ESI-MS). The method was demonstrated for six different organic compounds, analyzed individually. This is an interesting and potentially useful technique. The paper is quite short and lacking in important detail. There are a number of general and specific comments and questions that will need to be dealt with before it is acceptable for publication.

**General;**

The mass spectrometer apparently has very impressive resolution. Show us. Pick one or two sets of ions at a couple of nominal masses and show us a mass spectrum.

Authors: Thanks for your suggestion. We have addressed this in Question No. 24 in the Specific Question section below.

The materials of construction (tubing materials and dimensions) of the analytical system are not adequately described.

Authors: Thanks for your comment. We have addressed this comment in Question 19 of the Specific Questions section below.

There is at least one previous study that coupled and aerosol sampling method with ESI/MS (Stockwell et al., 2018) that showed that SMPS alone can have problems when relied on as the primary calibration method. Also, that methods relies on some assumptions in order derive the actual aerosol mass concentration. For example, are the authors assuming the generated particles are spheres with densities equal to the pure materials? Multiple charging can introduce errors, as noted by Stockwell et al. 2018. Did the authors consider that in their analysis of the SMPS?

Authors: Thanks for your comment and for providing relevant literature to our study. We have added this reference to the revised manuscript detailed in one of the responses to the specific question section. Assumptions of using SMPS were also added to the revised manuscript, and the details can be found in one of the responses in the specific question section. All SMPS data has corrected for multiple charging, which has minimal effect on our study.

**Specific;**

1.  Abstract – there could be a lot more detail here. Just tell us compounds that were used in the study, there are only six. Tell us the actual mass resolution that

was achieved in this study. Tell us what the limits of detection were, and how they were derived (1 sigma?).

Authors: Thanks for your suggestions. We have revised the abstract to include more information as follows:

Line 9 – 25

- **Abstract.** The chemical properties of aerosols in the atmosphere significantly influence their impact on the global climate forcing and human health. However, a real-time molecular-level characterisation of aerosols remains challenging due to the complex nature of their chemical composition. The current study constructed an instrumental system for the real-time chemical characterisation of aerosol particles. The proposed setup consists of a custom-built high-efficiency aerosol collector (HEAC) used to collect aerosol samples into a working fluid and an electrospray ionisation (ESI) Orbitrap Mass spectrometer (MS) for the subsequent chemical analysis of the liquid sample. The HEAC/ESI-Orbitrap-MS was calibrated against six chemical compounds—vanillic acid (VA), adonitol, erythritol, tricarballylic acid (TCA), sucrose, and trehalose—to investigate the system's sensitivity and limit of detection (LOD). Results showed that the coupled system has high sensitivities towards the tested chemical compounds and a similar, if not better, LOD than other related instrumental techniques. The 3 sigma LOD of the tested compounds ranged from $1.1 \pm 0.14$ ng m$^{-3}$ (erythritol) to $65 \pm 7.4$ ng m$^{-3}$ (TCA). The capability of the HEAC/ESI-Orbitrap-MS system to identify the chemical composition of organic aerosols (OA) was also examined. Sample OA was generated by α-pinene ozonolysis, and the chemical characterisation results were compared to similar studies. Our data showed that the HEAC/ESI-Orbitrap-MS system can identify most of the α-pinene ozonolysis products reported in the literature, including cis-pinonic acid, pinalic acid and 3-methyl-1,2,3-butanetricarboxylic acid (MBTCA). Monomeric and dimeric reaction products were accurately identified in the mass spectra, even at a total OA mass concentration $< 2$ μg m$^{-3}$. The present study showed that the HEAC/ESI-Orbitrap-MS system is a robust technique for the real-time chemical characterisation of OA particles under atmospheric relevant conditions.

2. Lines 38-48. One aspect missing from this is the fact that many sampling methods (e.g. filters) loose semi-volatile species. The authors seem unaware of the FIGAERO inlet work that couples short-term filters with HR-ToF-MS (Lopez-Hilfiker et al., 2014).

Authors: Thanks for your suggestion. We have added the sampling artifact problem of using filters in off-line analysis in the revised manuscript as follows:

Line 50 – 51

- Also, using filters for sample collection may result in positive sampling artifacts, as they may adsorb a portion of semi-volatile compounds in the sampled air (Kirchstetter et al., 2001).

We are aware of the FIGAERO inlet system for time-resolved chemical characterisation of aerosol samples. However, as an online instrument, its sampling cycle (over 30 mins) is significantly longer than the techniques described in the "real-time" system section. Therefore, categorising FIGAERO in this context does not really fit into the manuscript. Moreover, the introduction section aims to give an overview of online and offline-related instrumentation rather than a comprehensive review of all the existing techniques. Hence, we decided to omit FIGAERO from the introduction section.

3. Line 59. The ToF MS systems currently in commercial use can often provide adequate resolution at lower masses (<100 amu), they can struggle with complicated mixtures at higher masses.

Authors: Thanks for your suggestion. We have added this statement to the manuscript to make the content more comprehensive.

Line 62 - 64

- Additionally, although the TOF mass analyser used in both instruments can provide sufficient resolution for compounds with low molecular weight, they cannot unambiguously separate isobaric compounds in complex mixtures, making post-experiment data analysis challenging.

4. Lines 61-77. The authors appear unaware of the work of Stockwell et al., 2018, who coupled a PiLS system with negative ion electrospray (quadrupole) MS.

Authors: Thanks for your suggestion. We agree that this is an important literature describing a very similar technique to the current study. We have included the work of Stockwell et al., 2018 and revised the manuscript as follows:

Line 80 – 83

- Stockwell et al. (2018) have coupled PILS with an ESI quadrupole MS (PILS/ESI-QMS) in a direct infusion manner to obtain real-time information on reactive nitrogen species in aerosol samples. The proposed system in the current study has a similar configuration to the PILS/ESI-QMS, which consists of a custom-built aerosol collection system described in Brown et al. (2019) and coupled to a high-mass resolution mass spectrometer, namely Orbitrap MS.

5. Line 76 – 89. These sentences and paragraphs fragments are all mixed up and out of order. Lines 82-86 belong at the end of the Line 61-76 paragraph. The sentence on lines 76-77 and the other pieces in lines 78-89 should form the last paragraph of the introduction where you tell us what you will be presenting in this paper.

Authors: Thanks for your suggestions. We have revised the manuscript to improve the readability of this section as follows:

Line 78 – 96

- Therefore, it is necessary to construct a new system that can provide real-time determination of aerosol chemical composition at atmospheric-relevant mass loadings. Stockwell et al. (2018) have coupled PILS with an ESI quadrupole MS (PILS/ESI-QMS) in a direct infusion manner to obtain real-time information on reactive nitrogen species in aerosol samples. The proposed system in the current study has a similar configuration to the PILS/ESI-QMS, which consists of a custom-built aerosol collection system described in Brown et al. (2019) and coupled to a high-mass resolution mass spectrometer, namely Orbitrap MS.

  The high-efficiency aerosol collector (HEAC) achieves nearly 100% collection efficiency for fine and ultrafine particles down to 30 nm (Brown et al., 2019). The HEAC operates similarly to the particle-into-liquid sampler (PILS), using condensational growth of aerosol particles to enhance their collection efficiency. While PILS uses a washed-wall collector, HEAC employs a vortex collector—a cyclone with a standing liquid vortex on its wall during operation (Orsini et al., 2008). The Orbitrap MS used in the current study is a commercialised mass spectrometric technique that offers high mass resolution. In recent years, several high-resolution mass spectrometric techniques have been commercialised, allowing researchers to obtain detailed molecular information with minimal peak overlap. One such technique is the Orbitrap mass analyser coupled with an ESI source (Zubarev and Makarov, 2013). The ESI-Orbitrap-MS offers a mass resolution above 100,000, suitable for analysing complex mixtures containing numerous compounds, such as organic aerosol (OA) samples. Using an ESI source can also preserve the molecular identity of chemical compounds in the sample.

  The objective of the current study was to assess the performance of the newly constructed HEAC/ESI-Orbitrap-MS system. The system's sensitivity and detection limits were assessed using various chemical compounds, and its capability of providing real-time chemical information on aerosol samples was evaluated by analysing the products of α-pinene ozonolysis.

6. Methodology: This entire section was described without once telling us what polarity you were using for the ESI inlet.

Authors: Thanks for your reminder. We have added the corresponding information in the revised manuscript as follows:

Line 117 – 118

- The ESI-Orbitrap-MS was run in negative ion mode with a spray voltage of -3.5 kV.

7. Lines 118-120. You have skipped a step here. These are solids, so you must have used a solvent in order to nebulize them. What solvent and at what concentrations?

Authors: Thanks for your reminder. We have revised the manuscript to provide the information as follows:

Line 130 – 132

- Briefly, each examined compound was dissolved in around 50 mL of Milli-Q water and nebulised by an atomiser (Collison 1-jet, CH Technologies, USA). The nebulised sample was then pushed through a diffusion dryer filled with silica gel.

8. Line 123. This would be a good place to present all the assumptions that go into using the SMPS.

Authors: Thanks for your suggestion. Assumptions of calculating the mass concentration of different compounds were added to the revised manuscript as follows:

Line 135 – 136

- The mass concentration of the sample was calculated from the measured size distribution assuming that the particles are spherical and with the same density as their pure solid form.

9. Lines 124-129. Please show us some plots of MS signal versus time, what do those look like, the y-axis error bars in Figure 2 imply there is quite a bit of variability in some of them.

Authors: Thanks for your question. We have included a sample chromatogram obtained from the calibration trehalose in the supporting information as follows:

[Figure]

Figure S5. The chromatogram obtained from the trehalose calibration experiment. The blue arrows in the figure represent the time ranges used to calculate the averaged signal intensities and the standard deviations of the eight calibration points.

10. Line 156. How accurate is the assumed density and what factors might affect that?

Authors: Thanks for your question. The density of α-pinene ozonolysis products was estimated from high-resolution AMS and SMPS measurement data by Shiling et al., (2009). The calculated density will be affected by the chemical composition of the product and the relative abundance between different compounds within the product mixture. Shiling et al., observed a decrease in particle density with increasing α-pinene ozonolysis product mass loading. The density of α-pinene ozonolysis products used in the current study was taken from the Shilling et al., (2009) at a similar SOA mass loading.

11. Line 162. I can't tell what the phrase 'prove the functionalities' means, please explain.

Authors: Thanks for your comment. We have revised the statement in the manuscript to clarify the meaning as follows:

Line 175 – 176

- It should be noted that the α-pinene ozonolysis experiment carried out in the current study serves to assess the performance of the new HEAC/ESI-Orbitrap-MS system.

12. Section 2.3. Please show us a close-up of the mass separation achievable by the Orbitrap at one or two nominal masses.

Authors: Thanks for your question. We have included Figure S6 in the Supporting information to show the separation of peaks from m/z = 183.0000 to 183.1200 as follows:

[Figure]

Figure S6. The zoomed-in mass spectrum of Figure 5 from m/z = 183.0000 to 183.2000. Numbers above each peak correspond to the m/z of the detected ions. R represents the mass resolution of the detected peak, and the suggested chemical formula is given below.

13. Lines 190-192. Would inorganic salts create matrix effects that would alter sensitivities.?

Authors: Thanks for your question. Inorganic salts will likely cause ion suppression if present at high concentrations due to their high ionisation efficiencies in ESI. A note on the effect of inorganic salts on the sensitivity was added to the revised manuscript as follows:

Line 393 – 395

- However, in experiments with high OA mass loadings (e.g. over 200 µg m$^{-3}$) and using inorganic salts as seed particles, attention must be paid to the ion suppression and matrix effects if doing quantitative data analyses.

14. Figure 2. It seems inappropriate to extrapolate the fit line for VA way past the data points as it is done in this figure.

Authors: Thanks for your comment. We agree that some of the trend lines are unnecessarily extrapolated. Figure 2 was revised and replaced with the following figure:

[Figure]

15. Lines 233. It would be interesting to see the background signals that are being referred to.

Authors: Thanks for your question. The background signal here means the total ion signals, which is the TIC shown in figure 3.

16. Figure 4. The colors are hard to discern and are not color-blind compatible.

Authors: Thanks for your reminder. We have revised Figure 4 and increased the contrast of the colours used to make it more compatible for different readers as follows:

[Figure]

17. Figure 6. The designations ±H and ±O mean nothing to those not intimately familiar with mass defects. Please show us which direction is which and explain what they mean as far as H and O content go.

Authors: Thanks for your suggestions. We have added a description of how ±H and ±O should be read in Figure 6's caption as follows:

Line 287 – 289

- Dash lines in the figure represent the changes in mass defect when an oxygen (O) or hydrogen (H) atom was added to a compound. The addition of an H atom will increase the mass defect while the addition of an O atom will decrease the mass defect of a compound.

18. Line 281. What do you mean by 'the agreement between' are you talking about the agreement in the time profile?

Authors: Thanks for your reminder. We have revised the sentence to clarify the message we want to bring out as follows:

Line 298 – 300

- Nevertheless, the agreement between the time profile of the particle's mass concentration and the products' signal intensities confirmed the capability of the HEAC/ESI-Orbitrap-MS system to identify the chemical composition of organic aerosols in real time.

19. Line 299. Isn't there a limit in how small the inner diameter of tubing can be? It would help if you told us in the Methods Section what it is.

Authors: Thanks for your question. We have added details of the tubing we used in the Methodology section as follows:

Line 114 – 117

- Ethylene tetrafluoroethylene tubing (ETFE, 1/16" outer diameter, 0.04" inner diameter, IDEX Health & Science, USA) was used for the liquid flow lines upstream of the auxiliary pump. In contrast, polyether ether ketone tubing (PEEK, 1/16" outer diameter, 0.005" inner diameter, IDEX Health & Science, USA) was used to connect the auxiliary pump and the ESI source.

20. Figure 7. It would help in understanding this figure if you put shaded areas behind the time traces for each sampling mode.

Authors: Thanks for your suggestions. We have revised Figure 7 as follows:

[Figure]

21. Line 318. The 'apparent increase' is not very convincing. Maybe if you showed time averaged data.?

Authors: Thanks for your suggestions. We have revised the manuscript with averaged signal intensities to show the difference between the two phases as follows:

- The time profile of the dimer showed an increase in signal intensity from 2.29 ± 2.66 in the 3rd HEPA filter phase to 12.15 ± 5.26 in the fourth no-filter phase.

22. Lines 344-345. What is responsible for the 2-minute time constant? What is the residence time of the liquid in the vortex collector?

Authors: Thanks for your question. The 2-minute time constant was caused by the diffusion and dilution of the sample in the aqueous phase during the transportation of the sample from the vortex collector to the ESI source inlet. The volume of liquid in the vortex collector when it was in operation was around 100 µL (Orsini 2008). Given a flow rate of 1 mL min$^{-1}$, the residence time of the liquid in the vortex collector was around 10 s.

23. Lines 350-352. We have no context with which to evaluate these statements on improving the time response since we haven't been given the information of materials and dimensions of the tubing and we don't know what the time constant of the vortex collector is. It seems like the vortex would be something of a well-mixed element that would tend to smear out the signals in time.

Authors: Thanks for your question. Details of the material and dimension of the tubing were given in question 19, while the question regarding the vortex collector was addressed in question 22. Since the vortex collector is a fixed element in the setup and the tubing's ID was already small, we suggested optimising the system by shortening the length of the tubing in the manuscript.

24. Lines 385. What was the mass resolution (M/DM) for actual mass spectra?

Authors: Thanks for your question. Figure S6 was added to the supporting information to show the typical mass resolution of product peaks observed in the current study.

25. The reference list needlessly hard to read. Please put line breaks between citations or indent the first line of a citation

Authors: Thanks for your suggestion. We have put a line break between citations in the revised manuscript. We believe that the journal will have their own formatting requirements for the reference, and we will revise that accordingly.

26. SI – please put page numbers on.

Authors: Thanks for your reminder. Page numbers have been added to the revised Supporting Information accordingly.

27.   SI Eq1. It looks like you are quoting 1 sigma LODs? If true, that needs to be specified when you put those numbers in the abstract and elsewhere.

Authors: Thanks for your reminder. The calculation of the LOD was based on setting the S/N in equation (1) to 3, which means that the LODs reported in our study are 3 sigma LODs. We have added the description to the SI as follows:

Line 32 – 33 in SI

- A S/N of 3 was used to calculate the LODs in the current study.

28.   Table S1: Adonitol vapor pressure is 1.51 Pa and Erythritol vapor pressure is 6.36 ´ $10^{-4}$ Pa? I don't believe it, something is wrong. I checked my version of EPI suite and it says the same thing as the table, but it doesn't make sense if you look the two structures.

Authors: Thanks for your comment. We also believe that the estimation given by the EPI suite was not realistic. However, the EPI suite is the most readily available software, and it gives reasonably accurate estimations of the physical properties of different compounds most of the time. Therefore, we will keep the numbers in this table but add a note to remind the readers to pay attention to the uncertainties in these values.

Line 77 – 78 in SI

- It should be noted that the vapour pressures are model-estimated values. Cautions must be exercised when using these values in any kind of analysis.

29.   Table S4. This way of sorting the data, C#, is not very useful. I'd rather see it by exact mass. It would show where the clusters are (which would give you a good guide as to what nominal mass could be used to show the MS resolution). It also makes the mass defect O# count dependence clear.

Authors: Thanks for your comments and suggestions. We agree that sorting the compounds according to their exact masses is useful for investigating the MS resolution and observing the mass defects. However, we sort the α-pinene ozonolysis products according to their carbon number because it gives information on the structure of the compounds and the chemical processes they have undergone. For example, since α-pinene has a carbon number of 10, any products with less than 10 carbons correspond to C-C bond scission during the oxidation of α-pinene, while products with more than 10 carbons were probably resulting from the dimerization or aldol reaction of small (low carbon number) ozonolysis products. It's also easier to compare the products identified in the current study with those reported in the literature. Therefore, we would like to keep the table in the current format.

References:

Brown, R. A., Stevanovic, S., Bottle, S., and Ristovski, Z. D.: An instrument for the rapid quantification of PM-bound ROS: the Particle Into Nitroxide Quencher (PINQ), Atmos. Meas. Tech., 12, 2387-2401, 10.5194/amt-12-2387-2019, 2019.

Kirchstetter, T. W., Corrigan, C. E., and Novakov, T.: Laboratory and field investigation of the adsorption of gaseous organic compounds onto quartz filters, Atmospheric Environment, 35, 1663-1671, https://doi.org/10.1016/S1352-2310(00)00448-9, 2001.

Lopez-Hilfiker, F. D., Mohr, C., Ehn, M., Rubach, F., Kleist, E., Wildt, J., Mentel, T. F., Lutz, A., Hallquist, M., Worsnop, D., and Thornton, J. A.: A novel method for online analysis of gas and particle composition: description and evaluation of a Filter Inlet for Gases and AEROsols (FIGAERO), Atmos. Meas. Tech., 7, 983-1001, 2014.

Orsini, D. A., Rhoads, K., McElhoney, K., Schick, E., Koehler, D., and Hogrefe, O.: A Water Cyclone to Preserve Insoluble Aerosols in Liquid Flow—An Interface to Flow Cytometry to Detect Airborne Nucleic Acid, Aerosol Science and Technology, 42, 343-356, 10.1080/02786820802072881, 2008.

Stockwell, C. E., Kupc, A., Witkowski, B., Talukdar, R. K., Liu, Y., Selimovic, V., Zarzana, K. J., Sekimoto, K., Warneke, C., Washenfelder, R. A., Yokelson, R. J., Middlebrook, A. M., and Roberts, J. M.: Characterization of a catalyst-based conversion technique to measure total particle nitrogen and organic carbon and comparison to a particle mass measurement instrument, Atmos. Meas. Tech., 11, 2749-2768, 2018.

Zubarev, R. A. and Makarov, A.: Orbitrap Mass Spectrometry, Analytical Chemistry, 85, 5288-5296, 10.1021/ac4001223, 2013.

**Response to reviewer 2**

This manuscript presents an instrumental system combining a high-efficiency aerosol collector (HEAC) with ESI Orbitrap MS. This would be a potentially useful online instrument. While some descriptions are too much simplified, and some important details on the instrument are missing. The following comments need to be addressed:

1. The ionization efficiency of ESI is very sensitive to the sample matrix and ambient conditions. The authors also indicate these aspects in section 4.3. I may suggest to consider these effects more carefully.

(1) How to correct the variation of detected intensity caused by different sample matrix? If there are inorganic ions or organic molecules with high ionization efficiency, the ionization efficiency of one molecule would vary a lot. I agree that we cannot exclude the problems of ion suppression during the ionization of ESI, while the correction procedures need to be detailed here.

Authors: Thanks for your comment. We agree that the sample matrix, especially when inorganic ions or salts are present, will have a strong effect on the ionisation efficiencies of the target organics. Therefore, in section 4.3 of the manuscript, we suggested two possible ways of minimising ion suppression during the ESI process. We have elaborated it in more detail in the revised manuscript as follows:

Line 378 – 390

- Although it is impossible to eliminate the matrix effect when an ESI source is used, some correction and precautionary steps can be done to minimise or monitor its effects on the target compound's ionisation efficiency. For the proposed HEAC/ESI-Orbitrap-MS system, two practical ways of minimising and monitoring the matrix effect are to minimise the total concentration of compounds in the sample and add internal standards to the working fluid (Zhou et al., 2017). In particular, the total concentration of analytes should be kept below $10^{-5}$M in the current instrumental settings (Furey et al., 2013). This level corresponds to a mass concentration of around 100 μg m$^{-3}$ of soluble chemical components in the aerosol sample, assuming an average molecular mass of 200 for the soluble compounds. If the mass loading of the aerosol sample was too high, dilution of the sample should be considered. The sample dilution

can be accomplished by diluting the aerosol sample before entering the HEAC or infusing a larger amount of solvent into the aqueous sample flow before entering the ESI source. An internal standard can also be used to monitor the ion suppression effect during the sampling period. Ideally, for targeted analysis, stable isotope-labelled compounds should be used as they resemble the physical and chemical properties of the target analytes. However, isotope-labelled compounds might not be readily available for the target analytes. Alternatively, compounds with similar chemical functionalities to the target analytes that do not exist in the sample can be used as internal standards. The present study added malic acid to the working fluid as an internal standard in fast and slow α-pinene injection experiments.

(2) How about the stability of the instrument? I just wonder how much the detected intensity will change when ambient conditions, such as temperature, humidity, vary. For the measurement of ambient aerosol samples, ambient conditions are always changing.

Authors: Thanks for your question. Ambient conditions should have no effects on the performance of the instrument because all aerosol samples are passed through the condensation growth chamber in the HEAC, where they are exposed to supersaturated steam over 100 degrees. Therefore, the samples are "homogeneous" in terms of temperature and humidity before being collected by the working fluid in the vortex collector.

2. Six chemical compounds were used to Mass spectrometer equipped with ESI is quite good at analyzing organic molecules with -NO2, -OSO3 function groups, which has been widely used to quantify compounds such as nitro-aromatics, organosulfates, etc. I may suggest to test the sensitivity and LOD of these compound groups, at least test some commercial nitroaromatic compounds.

Authors: Thanks for your comment. We agree that the six chemicals tested in the current study didn't cover a wide enough spectrum of atmospheric-relevant compounds. However, the main objective of the current study is to prove that the coupling of the HEAC and the ESI-Orbitrap-MS can provide real-time information on the chemical characteristics of aerosol samples. The calibration and experimental data obtained in the current study were used to compare with other similar instrumental techniques to assess the performance of the proposed setup. Nevertheless, we agree that it is necessary to determine the

sensitivity and performance of the HEAC/ESI-Orbitrap-MS setup for other groups of compounds, including organosulfates and alkylnitrates, to gain a more comprehensive understanding of the aerosol's chemistry, and we will incorporate this into our future studies.

3. The author mentioned the possibility of thermal decomposition in ESI. Even if this cannot be concluded in the current study, some basis or evidence should be mentioned.

Authors: Thanks for your question. We agree that more details should be given to improve the comprehensiveness of the manuscript. We have revised the manuscript as follows:

Line 249 – 253

- Yang et al. (2018) investigated the thermal stability of xylitol and observed its thermal decomposition at around 250 °C. Given the structural similarity between xylitol and erythritol, it is possible that erythritol will undergo thermal decomposition at a similar temperature. However, since there is no reported thermal decomposition temperature of erythritol in the literature, the exact reason for the observed decrease in its signal intensity under high temperatures cannot be concluded in the current study.

References

Furey, A., Moriarty, M., Bane, V., Kinsella, B., and Lehane, M.: Ion suppression; A critical review on causes, evaluation, prevention and applications, Talanta, 115, 104-122, https://doi.org/10.1016/j.talanta.2013.03.048, 2013.

Yang, Y., Kong, W., and Cai, X.: Solvent-free preparation and performance of novel xylitol based solid-solid phase change materials for thermal energy storage, Energy and Buildings, 158, 37-42, https://doi.org/10.1016/j.enbuild.2017.09.096, 2018.

Zhou, W., Shuang, Y., and and Wang, P. G.: Matrix Effects and Application of Matrix Effect Factor, Bioanalysis, 9, 1839-1844, 10.4155/bio-2017-0214, 2017.

---

## Author Response (AR2)

**Response to Reviewer 1**

The authors have still not specified the concentrations of analytes in the aqueous solutions used in the particle nebulizer.

Authors: Thanks for the reviewer's reminder. The mass concentrations of analytes in the nebuliser were included in the revised manuscript as follows:

Line 131 – 132

- All examined compounds except adonitol had a mass concentration of around 0.5 g L$^{-1}$. The mass concentration of adonitol in the atomiser was around 3 g L$^{-1}$.

---

## Author Response (AR3)

**Response to editor**

Please make the following minor technical corrections prior to submitting a final version of the manuscript:
1) Please alter several of the figures by using different symbols for the different color lines. I tested a couple of your figures on an online colorblind test system and found that some of the curves were not distinguishable by people with red-green color blindness (deuteranopia), despite the use of reasonable color scales. Symbols are unambiguous.

Authors: Thanks for your reminder. We have checked the colour accessibility of all our figures and revised the colour scheme of Figures 2 – 4 and 7 in the manuscript and Figure S4 in the supporting information.

2) Please review the references and make sure they are consistent with Copernicus guidelines. For example, the titles of some of the references have capitalized first letters, while other titles do not. Reference manager type software always needs careful hand-checking and correction.

Authors: Thanks for your reminder. We have checked and revised the references in the manuscript and supporting information to make them comply with the journal's requirements.